# LEARNING FROM DEEP MODEL VIA EXPLORING LOCAL TARGETS

## ABSTRACT

Deep neural networks often have huge number of parameters, which posts challenges in deployment in application scenarios with limited memory and computation capacity. Knowledge distillation is one approach to derive compact models from bigger ones. However, it has been observed that a converged heavy teacher model is strongly constrained for learning a compact student network and could make the optimization subject to poor local optima. In this paper, we propose ProKT, a new model-agnostic method by projecting the supervision signals of a teacher model into the student's parameter space. Such projection is implemented by decomposing the training objective into local intermediate targets with approximate mirror descent technique. The proposed method could be less sensitive with the quirks during optimization which could result in a better local optima. Experiments on both image and text datasets show that our proposed ProKT consistently achieves the state-of-the-art performance comparing to all existing knowledge distillation methods.

## 1 INTRODUCTION

Advanced deep learning models have shown impressive abilities in solving numerous machine learning tasks (Devlin et al., 2018b; Radford et al., 2018; He et al., 2016). However, the advanced heavy models are not compatible with many real-world application scenarios due to the low inference efficiency and high energy consumption. Hence preserving the model capacity using fewer parameters has been an active research direction during recent years (Polino et al., 2018; Wu et al., 2016; Hinton et al., 2015). Knowledge distillation (Hinton et al., 2015) is an essential way in the field which refers to a model-agnostic method where a model with fewer parameters (student) is optimized to minimize some statistical discrepancy between its predictions distribution and the predictions of a higher capacity model (teacher).

Recently, it has been observed that employing a static target as the distillation objective would leash the effectiveness of the knowledge distillation method (Jin et al., 2019; Mirzadeh et al., 2019) when the capacity gap between student and teacher model is large. The underlying reason lies in common sense that optimizing deep learning models with gradient descent is favorable to the target which is close to their model family (Phuong & Lampert, 2019). To counter the above issues, designing the intermediate target has been a popular solution: Teacher-Assistant learning (Jin et al., 2019) shows that within the same architecture setting, gradually increasing the teacher size will promote the distillation performance; Route-Constrained Optimization (RCO) (Mirzadeh et al., 2019) uses the intermediate model during the teacher's training process as the anchor to constrain the optimization path of the student, which could close the performance gap between student and teacher model.

One reasonable explanation beyond the above facts could be derived from the perspective of curriculum learning (Bengio et al., 2009): the learning process will be boosted if the goal is set suitable to the underlying learning preference (bias). The most common arrangement for the tasks is to gradually increase the difficulties during the learning procedures such as pre-training (Sutskever et al., 2009). Correspondingly, TA-learning views the model with more similar capacity/model-size as the easier tasks while RCO views the model with more similar performance as the easier tasks, etc.

In this paper, we argue that the utility of the teacher is not necessarily fully explored in previous approaches. First, the intermediate targets usually discretize the training process as several periods and the unsmoothness of target changes in optimization procedure will hurt the very property of

introducing intermediate goals. Second, manual design of the learning procedure is needed which is hard to control and adapt among different tasks. Finally, the statistical dependency between the student and intermediate target is never explicitly constrained.

To counter the above obstacles, we propose ProKT, a new knowledge distillation method, which better leverages the supervision signal of the teacher to improve the optimization path of student. Our method is mainly inspired by the guided policy search in reinforcement learning (Levine & Koltun, 2013), where the intermediate target constructed by the teacher should be approximately projected on the student parameter space. More intuitively, the key motivation is to make the teacher model aware of the optimization progress of student model hence the student could get the "hand-on" supervision to get out of the poor minimal or bypass the barrier in the optimization landscape.

The main contribution of this paper is that we propose a simple yet effective model-agnostic method for knowledge distillation, where intermediate targets are constructed by a model with the same architecture of teacher and trained by approximate mirror descent. We empirically evaluate our methods on a variety of challenging knowledge distillation setting on both image data and text data. We find that our method outperforms the vanilla knowledge distillation approach consistently with a large margin, which even leads to significant improvements compared to several strong baselines and achieves state-of-the-art on several knowledge distillation benchmark settings.

## 2 RELATED WORK

In this section, we discuss several most related literature in model miniaturization and knowledge distillation.

**Model Miniaturization**. There has been a fruitful line of research dedicated to modifying the model structure to achieve fast inference during the test time. For instance, MobileNet (Howard et al., 2017) and ShuffleNet (Zhang et al., 2018a) modify the convolution operator to reduce the computational burden. And the method of model pruning tries to compress the large network by removing the redundant connection in the large networks. The connections are removed either based on the weight magnitude or the impact on the loss function. One important hyperparameter of the model pruning is the compression ratio of each layer. He et al. (2018) proposes the automatical tuning strategy instead of setting the ratio manually which are proved to promote the performance.

**Knowledge Distillation**. Knowledge distillation focuses on boosting the performance while the small network architecture is fixed. Hinton et al. (2015); Buciluǎ et al. (2006) introduced the idea of distilling knowledge from a heavy model with a relatively smaller and faster model which could preserve the generalization power. To this end, Buciluǎ et al. (2006) proposes to match the logits of the student and teacher model, and Hinton et al. (2015) tends to decrease the statistical dependency between the output probability distributions of the student model and the teacher model. And Zhang et al. (2018b) proposes the deep mutual learning which demonstrates that bi-jective learning process could boost the distillation performance. Orthogonal to output matching, many works have been conducted on matching the student model and teacher by enforcing the alignment on the latent representation (Yim et al., 2017; Jiao et al., 2019a; Sun et al., 2019). This branch of works typically involves prior knowledge towards the network architectures of student and teacher model which is more favorable to distill from the model with the same architecture. In the context of knowledge distillation, our method is mostly related to TA-learning (Mirzadeh et al., 2019) and the Route-Constraint Optimization(RCO) (Jin et al., 2019) which improved the optimization of student model by designing a sequence of intermediate targets to impose constraint on the optimization path. Both of the above methods could be well motivated in the context of curriculum learning, while the underlying assumption indeed varies: TA-learning views the increasing order of the model capacity implied a suitable learning trajectory; while RCO considers the increasing order of the model performance forms a favorable learning curriculum for student. However, there have been several limitations. For example, the sequence of learning targets that are set before the training process needs to be manually designed. Besides, targets are also independent of the states of the student which does not enjoy all the merits of curriculum learning.

**Connections to Other Fields**. Introducing a local target within the training procedure is a widely applied spirit in many fields of machine learning. Montgomery & Levine (2016) introduce the guided policy search where a local policy is then introduced to provide the local improved trajectory, which

has been proved to be useful towards bypassing the bad local minima. He et al. (2012) augmented the training trajectories by introducing the so called "coaching" distribution to ease the training burden and similarity. Le et al. (2016) introduce a family of smooth policy classes to reduce smooth imitation learning to a regression problem. Lu et al. (2018) introduce an intermediate target so-called mediator during the training of the auto-regressive language model, while the information discrepancy between the intermediate target and the model is constrained through the Kullback-Leibler(KL) divergence. Moreover, Chen et al. (2017) utilized the interpolation between the generator's output and target as the bridge to alleviate data sparsity and overfitting problems of MLE training. Expect from the distinct research communities and objectives, our method also differs from their methods in both the selection of intermediate targets, *i.e.* learned online versus designed by hands, and the theoretical motivation, *i.e.* the explicit constrain in mirror descent guarantee the good property on improvement.

## 3 METHODOLOGY

In this section, we first introduce the background of knowledge distillation and notations in Section 3.1. Then, in Section 3.2, we generalize and formalize the knowledge distillation methods with intermediate targets. In Section 3.3 we propose our method ProKT, and in Section 3.4 the connection between ProKT and mirror descent is demonstrated.

### 3.1 BACKGROUND ON KNOWLEDGE DISTILLATION

To start with, we introduce the necessary notations and backgrounds which are most related to our work. Taking an K-class classification task as an example, the inputs and label tuple is denoted as $(x, y) \in \mathcal{X} \times \mathcal{Y}$ and the label $y$ is usually in the format of a one-hot vector with dimension $K$. The objective in this setting is to learn a parameterized function approximator: $f(x; \theta) : \mathcal{X} \to \mathcal{Y}$. Typically, the function could be characterized as the deep neural networks. With the logits output as $u$, the output distribution $q$ of the neural network $f(x; \theta)$ could be acquired by applying the softmax function over the logits output $u$: $q_i = \frac{\exp(u_i/T)}{\sum_{j=1}^{K} \exp(u_j/T)}$, where $T$ corresponds to the temperature. The objective of knowledge distillation could be then written as:

$$\mathcal{L}_{\text{KD}}(\theta) = (1 - \alpha)H(y, q_s(\theta)) + \alpha T^2 H(p_t, q_s(\theta)). \tag{1}$$

Here $H$ denotes the cross entropy objective, *i.e.*, $H(p, q) = \sum_{i=1}^{K} -p_i \log q_i$ which is the KL divergence between $p$ and $q$ minus the entropy of $p$ (usually constant when $p = y$). $p_t$ is the output distribution of a given teacher model and $\alpha$ is the balanced weight between the standard cross entropy loss and the knowledge distillation loss from teacher. $T$ is the temperature. In the following formulations, we omit the $T$ by setting $T = 1$.

### 3.2 KNOWLEDGE DISTILLATION WITH DYNAMIC TARGET

In this section, we generalize and formalize the knowledge distillation methods with intermediate targets. We propose that previous knowledge distillation methods, either with a static target (i.e., the vanilla KD) or with hand-crafted discrete targets (i.e., Route-Constraint Optimization (RCO) (Jin et al., 2019)), cannot make full use of the knowledge from teacher. Instead, a dynamic and continuous sequence of targets is a better choice, and then we propose our method in the next section.

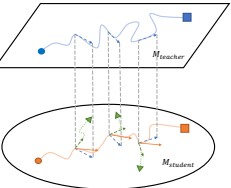

Figure 1: $\mathcal{M}_{teacher}$ and $\mathcal{M}_{student}$ refer to the output manifolds of student model and teacher model. The lines between circles (●,●) to squares (■,■) imply the learning trajectories in the distribution level. The intuition of ProKT is to avoid bad local optimas (triangles (▲)) by conducting supervision signal projection.

Firstly, we generalize and formalize the knowledge distillation methods with intermediate targets, named as *sequential optimization knowledge distillation* (SOKD) methods. Instead of conducting a static teacher model in vanilla KD, the targets to the student model of SOKD methods are changed during the training time. Without loss of generality, we denote the sequence of intermediate target distributions as $P_t = [p_t^1, p_t^2, \cdots, p_t^m, \cdots]$. Starting from a random initialized parameters $\theta^0$, the student model is optimized by gradient descent methods to mimic its intermediate target $p_t^m$:

$$\theta^m = \theta^{m-1} - \beta \nabla_\theta \mathcal{L}^m(\theta^{m-1}), \tag{2}$$

$$\mathcal{L}^m(\theta) = (1-\alpha)H(y, q_s(\theta)) + \alpha H\left(p_t^m, q_s(\theta)\right). \tag{3}$$

One choice to organize the intermediate targets is to split the training process into intervals and adopt a fixed target in each intervals, named as discrete targets. For example, the Route-Constraint Optimization (RCO) (Jin et al., 2019) saves the un-convergent checkpoints of teacher during the teacher's training to construct the target sequence. The learning target of student is changed every few epochs.

However, the targets are changed discontinuously in the turning points between discrete intervals, which would incur negative effects on the dynamic knowledge distillation. Firstly, switching to a target that is too difficult for the student model would undermine the advantages of curriculum learning. If the target is changed sharply to a model with large complexity improvement, it is hard for student to learn. Besides, the ineligible gap between adjacent targets would make the training process unstable and hurt the convergence property during the optimization (Zhou et al., 2018).

Therefore, we propose to replace the discrete target sequence with a continuous and dynamic one, whose targets are adjusted smoothly and dynamically according to the status of student model. In continuous target sequence, targets in each step are changed smoothly with ascending performance. In that case, if the student learns the target well in current step, the target of the next step is easier to learn because of the slight performance gap. The training process is stable as well, because the training targets are improved smoothly. Specifically, the optimization trajectories of the teacher model naturally offer continuous supervision signals for the student. In our work, we propose to conduct the optimization trajectories of teacher model as the continuous targets. Besides, to ensure that intermediate teachers are kept easy to learn for students, we introduce an explicit constraint in the objective of the teacher. This constraint dynamically adjusts the updating path of the teacher according to learning progress of the student. The key motivation of our method is illustrated in Fig. 1.

### 3.3 PROXIMAL KNOWLEDGE TEACHING

In this section, we firstly propose the SOKD adopting the optimization trajectories of teacher as the continuous targets. The learning process is that every time the teacher model updates one step towards the ground-truth, the student model updates one step towards the new teacher. Then based on this, we propose the *Proximal Knowledge Teaching* (ProKT), which modifies the updating objective of the teacher by explicitly constraining it in the neighbourhood of student model.

To construct the target sequence with continuous ascending target distributions, a natural selection is the gradient flow of the optimization procedure of the teacher distribution. With the student $q_{\theta_s}$ and teacher model $p_{\theta_t}$ initialized at the same starting point (e.g., $q_{\theta_s^0}(y|x) = p_{\theta_t^0}(y|x) = Uniform(1, K)$), we iteratively update the teacher model and the student model according to the following loss functions:

$$\theta_t^{m+1} = \theta_t^m - \eta_t \nabla \mathcal{L}_t(\theta_t^m), \quad \mathcal{L}_t(\theta_t) = H(y, p_{\theta_t}), \tag{4}$$

$$\theta_s^{m+1} = \theta_s^m - \eta_s \nabla \mathcal{L}_s(\theta_s, p_{\theta_t^{m+1}}), \quad \mathcal{L}_s(\theta_s) = H(p_{\theta_t}, q_{\theta_s}). \tag{5}$$

Here, the $\eta_t$ and $\eta_s$ are learning rates of student and teacher models, respectively. Starting with the same initialized distribution, the teacher model is updated firstly by running a step of stochastic gradient descent. Then, the student model learns from the updated teacher model. In this process, the student could learn from the optimization trajectories of the teacher model, which provides the knowledge of how the teacher model is optimized from a random classifier to a good approximator. Compared with the discrete case such as RCO, the targets are improved progressively and smoothly.

---

**Algorithm 1** ProKT

1: **Input:** Initialized student model $q_{\theta_s}$ and teacher model $q_{\theta_t}$. Data set $\mathcal{D}$.
2: **while** not converged **do**
3:     Sample a batch of input $(x, y)$ from the dataset $\mathcal{D}$.
4:     update teacher by $\theta_t \leftarrow \theta_t - \eta_t \nabla_{\theta_t} \hat{\mathcal{L}}_{\theta_t}$.
5:     update student by $\theta_s \leftarrow \theta_s - \eta_s \nabla_{\theta_s} \mathcal{L}(\theta_s)$.
6: **end while**

---

However, simply conducting iterative optimization following Eq. 4 with gradient descent could not guarantee the teacher would stay close to the student model even with a small update step. The gradient descent step of teacher in Eq. 4 is equivalent to solving the following formulation:

$$\theta_t^{m+1} = \arg\min_\theta \mathcal{L}(\theta_t^m) + \nabla_\theta \mathcal{L}(\theta)^\top (\theta - \theta_t^m) + \frac{1}{2}\eta_t \|\theta - \theta_t^m\|^2,$$

which only seeks the solution in the neighborhood of current parameter $\theta_t^m$ in terms of the Euclidean distance. Unfortunately, there is no explicit constraint that the target distribution $p_{\theta_t^{m+1}}(y|x)$ stays close to $p_{\theta_t^m}(y|x)$. Besides, because the learning process of teacher model is ignorant of how the student model has been trained, it is probably that the gap between student model and teacher model grows cumulatively.

Therefore, in order to constrain the target distribution to be easy-to-learn for the student, we modify the training objective of teacher model in Eq. 4 by explicitly bounding the KL divergence between the teacher distribution and student distribution:

$$\theta_t^{m+1} = \min_{\theta_t} H(y, p_{\theta_t}) \quad \text{s.t. } D_{\text{KL}}(q_{\theta_s}^m, p_{\theta_t}) \leq \epsilon. \tag{6}$$

The $\epsilon$ controls the how close the teacher model for the next step to the student model. In this case, we make an approximation that if the KL divergence of target distribution and the current student distribution is small, this target is easy for student to learn. By optimizing the Eq. 6, the teacher is chosen as the best approximator of the teacher model's family in the neighbour of student distribution.

With slight variant of the Lagrangian formula of Eq. 6, the learning objective of teacher model in ProKT is

$$\hat{\mathcal{L}}_{\theta_t} = (1 - \lambda)H(y, p_{\theta_t}) + \lambda H(q_{\theta_s}, p_{\theta_t}), \tag{7}$$

in which the hyper-parameter $\lambda$ controls the difficulty of teacher model compared with student model. The overall algorithm is summarized in Algorithm 1.

### 3.4 PROKT AS APPROXIMATE MIRROR DESCENT

Following the assumption that supervised learning could globally solve a convex optimization problem, it could be shown the proposed method corresponds to a special case of mirror descent (Beck & Teboulle, 2003) with the objective as $H(y, q_{\theta_s})$. Note the optimization procedure is conducted on the output distribution space, the constraint is the solution must lie on the manifold of output distributions which could be characterized in the same way as the student model. We use $\mathcal{Q}_{\theta_s}$ to denote the possible output distribution family with the same parameterization as the student model.

**Proposition 1** *The proposed ProKT solves the optimization problem:*

$$q_{\theta_s} \leftarrow \arg\min_{q_{\theta_s} \in \mathcal{Q}_{\theta_s}} H(y, q_{\theta_s})$$

*with mirror descent by iteratively conducting the following two step optimization at step $m$:*

$$q_{\theta_t}^m \leftarrow \arg\min_{q_{\theta_t}} H(y, q_{\theta_t}) \text{ s.t. } D_{KL}\left(q_{\theta_s}^m, q_{\theta_t}^m\right) \leq \epsilon, \quad q_{\theta_s}^{m+1} \leftarrow \arg\min_{q_{\theta_s} \in \mathcal{Q}_{\theta_s}} D_{KL}\left(q_{\theta_t}^m, q_{\theta_s}\right) \tag{8}$$

The first step is to find a better output distribution which minimizes the classification task and is close to the previous student distribution $q_{\theta_s}^m$ under the KL divergence. While the second step projects the distribution in the distribution family $\mathcal{Q}_{\theta_s}$ in terms of the KL divergence. The monotonic property directly follows the monotonic improvement in mirror descent (Beck & Teboulle, 2003).

## 4 EXPERIMENTS

### 4.1 SETUP

In order to evaluate the performance of ProKT under different knowledge distillation settings, we implement the ProKT in different tasks (image recognition and text classification), different network architectures, and different training objectives.

#### 4.1.1 IMAGE RECOGNITION

The image classification experiments are conducted in CIFAR-100 (Krizhevsky et al., 2009) following Tian et al. (2019).

**Settings**. Following the Tian et al. (2019), we compare the performance of knowledge distillation methods under various architecture of teacher and student models. We use the following models as teacher or student models: vgg (Simonyan & Zisserman, 2014), MobileNetV2 (Sandler et al., 2018) (with a width multiplier of 0.5), ShuffleNetV1 (Zhang et al., 2018a), ShuffleNetV2 (Simonyan & Zisserman, 2014), Wide Residual Network (WRN-$d$-$w$) (Zagoruyko & Komodakis, 2016) (with depth $d$ and width factor $w$) and ResNet (He et al., 2016). To evaluate the ProKT under different distillation loss, we conduct the ProKT with standard KL divergence loss and contrastive representation distillation loss proposed by CRD (Tian et al., 2019).

**Baselines**. We compare our model with the following baselines: vanilla KD (Hinton et al., 2015), CRD (Tian et al., 2019) and RCO (Jin et al., 2019). Results of baselines are from the report of Tian et al. (2019), except for the RCO (Jin et al., 2019), which is implemented by ourselves.

#### 4.1.2 TEXT CLASSIFICATION

Text classification experiments are conducted following the setting of (Turc et al., 2019) and (Jiao et al., 2019b) on the GLUE (Wang et al., 2018) benchmark.

**Datasets**. We evaluate our method for sentiment classification on SST-2 (Socher et al., 2013), natural language inference on MNLI (Williams et al., 2017) and QNLI (Rajpurkar et al., 2016), and paraphrase similarity matching on MRPC (Dolan & Brockett, 2005) and QQP[1].

**Settings**. The teacher model is the BERT-base (Devlin et al., 2018a) fine-tuned in the training set, which is a 12-layer Transformers (Vaswani et al., 2017) with 768 hidden units. Following the setting of Turc et al. (2019) and Jiao et al. (2019b), a BERT of 6 layer Transformers and 786 hidden units is conducted as the student model. For distillation between heterogeneous architectures, we use a single-layer bi-LSTM with 300 embedding size and 300 hidden size as student model. We implement the basic ProKT with standard KL divergence loss, and combine our method with the TinyBERT (Jiao et al., 2019b) by replacing the second stage of fine-tuning TinyBERT with our ProKT. More experimental details are listed in the supplementary materials.

**Baselines**. We compare our method with following baselines: (1) BERT + Finetune, fine-tune the BERT student on training set; (2) BERT/bi-LSTM + KD, fine-tune the BERT student or train the bi-LSTM on training set using the vanilla knowledge distillation loss (Hinton et al., 2015); (3) Route Constrained Optimization (RCO) (Jin et al., 2019), use 4 un-convergent teacher checkpoints as intermediate training targets; (4) bi-LSTM: train bi-LSTM in training set; (5) TinyBERT (Jiao et al., 2019b): match the attentions and representations of student model with teacher model on the first stage and then fine-tune by the vanilla KD loss on the second stage. For vanilla KD methods, we set the temperature as 1.0 and only use the KL divergence with teacher outputs as loss. We also compare our method with the results reported by Sun et al. (2019) and Turc et al. (2019).

### 4.2 RESULTS

Results of image classification on CIFAR100 are shown in Tab. 1. The performance is evaluated by top-1 accuracy. Results of text classification are shown in Tab. 2. The accuracy or f1-score on test set are obtained by submitting to the GLUE (Wang et al., 2018) website. Results on both text and image classification tasks show that ProKT achieves the best performance under almost all model settings.

---

[1]https://data.quora.com/First-Quora-Dataset-Release-Question-Pairs.

Table 1: Top-1 test *accuracy* (%) of student networks distilled from teacher with different network architectures on CIFAR100. Results except the RCO, ProKT and CRD+ProKT are from Tian et al. (2019).

| Teacher
Student | vgg13
MobileNetV2 | ResNet50
MobileNetV2 | ResNet50
vgg8 | resnet32x4
ShuffleNetV1 | resnet32x4
ShuffleNetV2 | WRN-40-2
ShuffleNetV1 |
|---|---|---|---|---|---|---|
| Teacher | 74.64 | 79.34 | 79.34 | 79.42 | 79.42 | 75.61 |
| Student | 64.6 | 64.6 | 70.36 | 70.5 | 71.82 | 70.5 |
| KD* | 67.37 | 67.35 | 73.81 | 74.07 | 74.45 | 74.83 |
| RCO | 68.42 | 68.95 | 73.85 | 75.62 | **76.26** | 75.53 |
| ProKT | **68.79** | **69.32** | **73.88** | **75.79** | 75.59 | **76.02** |
| CRD | 69.73 | 69.11 | 74.30 | 75.11 | 75.65 | 76.05 |
| CRD+KD | **69.94** | 69.54 | 74.58 | 75.12 | 76.05 | 76.27 |
| CRD+ProKT | 69.59 | **69.93** | **75.14** | **76.0** | **76.86** | **76.76** |

Table 2: Test results of different knowledge distillation methods in GLUE.

| Model | SST-2
(acc) | MRPC
(f1/acc) | QQP
(f1/acc) | MNLI
(acc m/mm) | QNLI
(acc) |
|---|---|---|---|---|---|
| BERT$_{12}$ (teacher) | 93.4 | 88.0/83.2 | 71.4/89.2 | 84.3/83.4 | 91.1 |
| PF (Turc et al., 2019) | 91.8 | 86.8/81.7 | 70.4/88.9 | 82.8/82.2 | 88.9 |
| Sun et al. (2019) | 92.0 | 85.0/79.9 | 70.7/88.9 | 81.5/81.0 | 89.0 |
| BERT$_6$ + Finetune | 92.6 | 86.3/81.4 | 70.4/88.9 | 82.0/80.4 | 89.3 |
| BERT$_6$ + KD | 90.8 | 86.7/81.4 | 70.5/88.9 | 81.6/80.8 | 88.9 |
| BERT$_6$ + RCO | 92.6 | 86.8/81.4 | 70.4/88.7 | 82.3/81.2 | 89.3 |
| BERT$_6$ + ProKT ($\lambda = 0$) | 92.9 | **87.1/82.3** | 70.7/88.9 | 82.5/81.3 | 89.4 |
| BERT$_6$ + ProKT | **93.3** | 87.0/82.3 | **70.9/88.9** | **82.9/82.2** | **89.7** |
| TinyBERT$_6$ (Jiao et al., 2019b) | 93.1 | 87.3/82.6 | **71.6/89.1** | **84.6**/83.2 | 90.4 |
| TinyBERT$_6$ + ProKT | **93.6** | **88.1/83.8** | 71.2/89.2 | 84.2/**83.4** | **90.9** |
| bi-LSTM | 86.3 | 76.2/67.0 | 60.1/80.7 | 66.9/66.6 | 73.2 |
| bi-LSTM + KD | 86.4 | 77.7/68.1 | **60.7/81.2** | 68.1/67.6 | 72.7 |
| bi-LSTM + RCO | 86.7 | 76.0/67.3 | 60.1/80.4 | 66.9/67.6 | 72.5 |
| bi-LSTM + ProKT ($\lambda = 0$) | 86.2 | 80.1/71.8 | 59.7/79.7 | 68.4/68.3 | 73.5 |
| bi-LSTM + ProKT | **88.3** | **80.3/71.0** | 60.2/80.4 | **68.8/69.1** | **76.1** |

Results show that the continuous and dynamic targets are helpful to take advantage of the knowledge from the teacher. Although adopting discrete targets in RCO could improve the performance to vanilla KD, our ProKT with continuous and dynamic targets is more effective in teaching student. To further show the effectiveness of continuity and adaptiveness (i.e., the KL divergence term to student in the update of teacher) in ProKT respectively, we test the results of ProKT with $\lambda = 0$, in which the targets are improved smoothly but without the adjustment towards the student. As shown in Tab. 2, the continuous targets are better than discrete targets (i.e., RCO), while incorporating the constraint from student when updating teacher could further improve the performance.

ProKT is effective as well when it is combined with different objective of knowledge distillation. When combined with contrastive representation learning loss in CRD, as shown in Tab. 1, and combined with TinyBERT in Tab. 2, ProKT could further boost the performance and achieves the state-of-the-art results in almost all settings.

ProKT is especially effective when the student is of different structure with teacher. As shown in Tab. 2, when the student is bi-LSTM, directly distilling knowledge from a pre-trained BERT has a minor effect. ProKT could improve a larger margin for bi-LSTM than small BERT when distilled from BERT-base. Since learning from a heterogeneous teacher is more difficult, exposing teacher's training process to student could offer better guidance to the student.

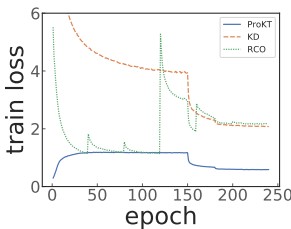 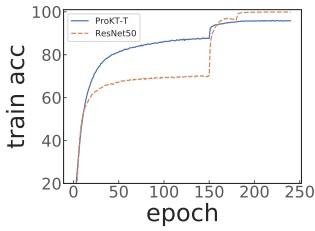

(a) Train loss          (b) Train accuracy

Figure 2: Training loss and accuracy for MobileNetV2 distilled from ResNet50 on CIFAR 100.

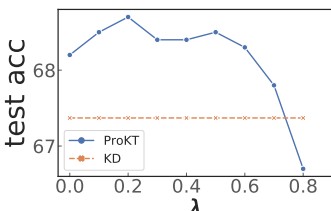 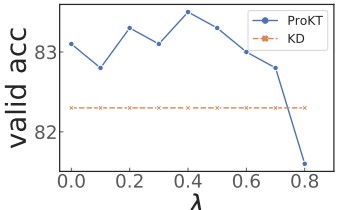

(a) VGG13 to MobileNetV2 in CIFAR 100.    (b) BERT$_{12}$ to BERT$_6$ in MNLI-mm.

Figure 3: Test/valid accuracy with different value of $\lambda$ for teacher and student model in ProKT.

### 4.3 DISCUSSION

#### 4.3.1 TRAINING DYNAMICS

To visualize the training dynamics of teacher model and student model, we show the training loss of student model and the training accuracy of teacher model in Fig. 2. The training losses are calculated by the KL divergence between the student model and their intermediate targets. Fig. 2a shows that the divergence between student and teacher in ProKT (i.e., the training loss for ProKT) is smooth and well bounded to a relative small value. For discrete targets in RCO, the divergence is bounded well in the beginning of training. However, at the target switching points, there are impulses in the training curve and then the loss is kept to a relative larger value.

Then, we examine the performance of teacher model in Fig. 2b. ResNet50 refers to the teacher model which is trained by vanilla loss. While the ProKT-T denotes the teacher model which updated by the ProKT loss. It could be found that the performance of teacher model in ProKT deteriorates because of the "local" constraint from student. However, the lower training accuracy for teacher model does not affect the training performance of the student model as illustrated in the Tab. 1 which further justified our intuition of involing local targets.

#### 4.3.2 ABLATION STUDY

To test the impact of the constraint from student in Eq. 6, test and valid accuracy with respect to different $\lambda$ for image and text classification tasks are shown in Fig. 3. It is illustrated that the performance is improved in an appropriate range of $\lambda$, which means that the constraint term is helpful to provide appropriate targets. However, when the $\lambda$ is too large, the regularization from student will heavily damage the training of teacher and the performance of student will drop.

### 5 CONCLUSION

We propose a novel model agnostic knowledge distillation method, ProKT. The method projects the step-by-step supervision signal on the optimization procedure of student with an approximate mirror descent fashion, *i.e.*, student model learns from a dynamic teacher sequence while the progressive teacher is aware of the learning process of student. Experimental results show that ProKT achieves good performance in knowledge distillation for both image and text classification tasks.

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

# A   APPENDIX

## A.1   EXPERIMENTAL DETAILS FOR TEXT CLASSIFICATION

We use the pre-trained BERTs released by Turc et al. (2019) except for TinyBERTs. For TinyBERTs, we use the pre-trained model released by Jiao et al. (2019b)[2]. We fine-tune 4 epoch for non-distillation training and 6 epoch for distillation training. Adam (Kingma & Ba, 2014) optimizer with learning rate 0.001 is used for biLSTM and with a learning rate from {3e-5, 5e-5, 1e-4} is used for BERTs. The hyper-parameter of $\lambda$ in Eq. 6 is chosen according to the performance in the validation set. For ProKT in TinyBERT, we use the data argumentation following Jiao et al. (2019b).

## A.2   FULL COMPARISON OF KD IN IMAGE RECOGNITION SEC EXPERIMENT RESULTS OF HOMOGENEOUS ARCHITECTURE KD IN IMAGE RECOGNITION

We provide the full comparison of our method with respect to several additional knowledge distillation methods as extension in the Table. 3.

Table 3: Top-1 test *accuracy* (%) of student networks distilled from teacher with different network architectures on CIFAR100. Results except the RCO, ProKT and CRD+ProKT are from Tian et al. (2019).

| Teacher
Student | vgg13
MobileNetV2 | ResNet50
MobileNetV2 | ResNet50
vgg8 | resnet32x4
ShuffleNetV1 | resnet32x4
ShuffleNetV2 | WRN-40-2
ShuffleNetV1 |
|---|---|---|---|---|---|---|
| Teacher | 74.64 | 79.34 | 79.34 | 79.42 | 79.42 | 75.61 |
| Student | 64.6 | 64.6 | 70.36 | 70.5 | 71.82 | 70.5 |
| KD* | 67.37 | 67.35 | 73.81 | 74.07 | 74.45 | 74.83 |
| FitNet* | 64.14 | 63.16 | 70.69 | 73.59 | 73.54 | 73.73 |
| AT | 59.40 | 58.58 | 71.84 | 71.73 | 72.73 | 73.32 |
| SP | 66.30 | 68.08 | 73.34 | 73.48 | 74.56 | 74.52 |
| CC | 64.86 | 65.43 | 70.25 | 71.14 | 71.29 | 71.38 |
| VID | 65.56 | 67.57 | 70.30 | 73.38 | 73.40 | 73.61 |
| RKD | 64.52 | 64.43 | 71.50 | 72.28 | 73.21 | 72.21 |
| PKT | 67.13 | 66.52 | 73.01 | 74.10 | 74.69 | 73.89 |
| AB | 66.06 | 67.20 | 70.65 | 73.55 | 74.31 | 73.34 |
| FT* | 61.78 | 60.99 | 70.29 | 71.75 | 72.50 | 72.03 |
| NST* | 58.16 | 64.96 | 71.28 | 74.12 | 74.68 | 74.89 |
| RCO | 68.42 | 68.95 | 73.85 | 75.62 | **76.26** | 75.53 |
| ProKT | **68.79** | **69.32** | **73.88** | **75.79** | 75.59 | **76.02** |
| CRD | 69.73 | 69.11 | 74.30 | 75.11 | 75.65 | 76.05 |
| CRD+KD | **69.94** | 69.54 | 74.58 | 75.12 | 76.05 | 76.27 |
| CRD+ProKT | 69.59 | **69.93** | **75.14** | **76.0** | **76.86** | **76.76** |

---

[2]https://github.com/huawei-noah/Pretrained-Language-Model/tree/master/TinyBERT

