# OpenReview forum: "Learning from deep model via exploring local targets"
_ICLR.cc/2021/Conference — Reject_

### Official Review · AnonReviewer1 · 2020-10-25
**comments of step-wise distillation**

**Rating:** 4
**Confidence:** 5

**Review:**

Overview summary:

This submission studies the knowledge distillation approach for deep learning models. The aspect is focused on the intermediate target models when learning for the student models. The authors claimed the previous teacher targets are poor to make student model to local optima or be unstable. The authors propose a new "model-agnostic" method, named ProKT, proximal knowledge teaching, which is implemented by decomposing the training objective of the student model into several local intermediate targets with approximate mirror descent technique. Specifically, the training of the teacher model and student model is step-wise, which means the teacher model updates and distills the knowledge after one step, and the student model then takes the teacher model at the current step as a target, and train the student model itself. To make the training more stable or less sensitive, the teacher model is constrained by a gap bounding between the teacher output distribution and student output distribution (KL divergence). The authors then evaluate their method on image classification task (CIFAR-100) and several text classification tasks, under different model structures, different training objectives. The results show that ProKT is able to achieve effective performance compared to previous knowledge distillation methods.

General comments:
Overall speaking, the motivation of this submission is clearly described and the idea is easy to follow. I understand the authors are making a better job towards the distillation approach. But after carefully reading the method and the experimental results, I do have several concerns, which make me feel things can be better improved or better modeled. Please refer to my following detailed comments on the pros and cons.

Pros:
1. First, I personally like the idea of better studying knowledge distillation problem, since kd is a very important and effective method in current deep learning models. This submission studies the internal learning path of the student model from the teacher model, with the viewpoint of how to modify the teacher model instead of the student model. The main claim is about the teacher targets (multiple intermediate targets) should be continuous so that the student model can be more stable to follow the guided signals. This is a good point.
2. The authors present a clear description of the motivation and the proposed approach, which is very easy to follow. As for the implementation, it seems easy to do, though the authors do not open source their implementation.
3. The authors conduct experiments on two different domains, which are the image and text classifications. The verification is performed under different settings (models, training objectives) so to demonstrate the universal usefulness of the approach.

Cons:
Though the interesting points and the general well modeling. I do feel several concerns as follows:
1. To be honest, the proposed method mainly do a step-wise distillation for the student model. To make a better constraint and stable training, the teacher model is constrained with the learning of the student model through a KL divergence between the two output distributions. This contribution is a little bit limited. First, the step-wise distillation is straight forward, the authors mainly criticize the discrete targets from previous methods, but the continuous setting modeled in this paper is not a very elegant way or surprising way. This modeling would cost several problems (as I listed in the below comments). This method is also very similar to mutual training, instead of the teacher and student models are different architectures. Second, the constraint between the student model and teacher model is somehow hard to make a clear convincing claim. In my opinion, the teacher model aims to make a strong performance by learning from the supervised target $y$, the goal mainly targets at the performance, this constraint may somehow limit the performance of the strong teacher model (e.g., figure 2(b)). Though the teacher model wants to provide a continuous signal for the student model (as the author claimed), the "step-wise learning/distillation" can guarantee this assumption in most cases for the student model. The probable large gap is not very possible (if the author can give several strong examples). Therefore, this is not so convincing.
2. The continuous targets distillation have some problems from my side. For example, the training cost is increased for the student model. Previous knowledge distillation methods usually do several times distillation or only one-time distillation, which should be efficient to train since the student model is smaller than the teacher model, therefore the training step is less and the convergence speed is faster. With the step-wise distillation, the update steps are increased to be the same as the teacher model.  Besides, for ProKT, the memory (e.g., internal backpropagation parameters) request is more than previous methods, since the approach should update the teacher model and student model at the same time. While for previous works, only the update for the student model parameters required.
3. As for the experiments, in Table 1, the gap between RCO and ProKT is really limited, which makes it hard to say that ProKT is stronger or more effective than RCO. The performances are similar, only slightly better from ProKT. A more confused setting is in Table 2, the BERT$_6$ + ProKT ($\lambda=0$) is similar to the next line, which means the constraint is not so important (though in bi-LSTM, it shows better performance), or at least not stable/universally work (somehow this verifies my point of the small gap between teacher and student model distributions of my point 2). This seems to be a negative point for the verification. The differences between the bi-LSTM and BERT experiments also should be better claimed.

Several other points and questions:
1. The detailed values for $\lambda$ and $\alpha$ should be reported, in order to give a clear setting.
2. In figure 2(a), the bleu line fo the student model training is interesting, it first increases the loss, do the authors have more explanations?
3. In figure 3(a), the better way is to show the valid acc instead of test acc, this would be fairer.
4. Minor errors, in 4.3.1, the third line of the second phrase, "It could be found hat xxx" -> "It could be found that xxx".
5. A suggestion is that the detailed parameter number should be reported for the teacher and student models.

-------------------
updated comments:
I thank the authors to provide responses before the last time of rebuttal. Some of the questions are addressed, but it seems many points still keep disagreement. For example, the contribution is still not clear to me The training cost is increased a lot, this can not be ignored, though I understand in the industry it may be less important. However, for academic research, we should care about this. Besides, it seems that all reviewers agree with the limited contribution and unsatisfied experiments.

---

> ### Author Response · Authors · 2020-11-24
> **Thanks for your insightful comments!**
>
> Thanks for your insightful comments! Following are responses to your questions.
>
> ### Q: the contribution is limited.
> The contribution of our method includes two folds. First, we replace the discrete targets with continuous targets, which are constrained by the KL divergence to student distribution. Furthermore, we theoretically related our iterative optimization process with a mirror descent optimization, which reveals the insight of this model as projecting the teacher distributions to the distribution space of student models. As far as we know, other works such as mutual learning do not give such insight.
>
> The constraint for teacher training is essential in our approach, for it represents the projection of the updated teacher distribution to the neighborhood of student distribution. For knowledge distillation, a teacher model with stronger performance does not always induce a better student, which has been proven empirically by a series of work, such as TA learning and RCO. Instead, a teacher providing easy-to-learn targets for students is beneficial. Thus, we introduce an explicit constraint for teacher training to guarantee the target distributions close to student distributions. Without this explicit constraint, the direction of parameters update is only determined by its cross-entropy with groud-truth, and it might cause the updated distribution to deviate heavily from the directions which are aligned with student distribution.
>
> ### Q: comparison with mutual learning
> In approach, the key difference between our method and mutual learning is that prokt has an **asymmetric** training objective: the student model does not get the supervision signal from the ground truth label while the teacher model does which corresponds to our theoretical motivation of gradient projection.
>
> ### Q: training cost for ProKT
> The target of knowledge distillation is to improve the efficiency of inference by using smaller and faster models. The training cost is not the main concern for the deployment of deep learning models.
>
> ### Q: the gap between RCO and ProKT is limited and the differences between the bi-LSTM and BERT experiments also should be better claimed
> A: The improvement of ProKT over RCO is consistent and reliable. Furthermore, when the student model is heterogeneous to the teacher model (e.g. BERT_base to biLSTM), the advantage of our method will be more significant, because ProKT is able to reduce the capacity gap between teacher and student by projecting the signal of the teacher model into the function space of the student model. As shown in Table.2, improvement between heterogeneous models (e.g. BERT_base to biLSTM) is larger than improvement with homogeneous models (e.g. BERT_base to BERT_6).
>
> ###  Other points:
>
> **Q: detailed values for lambda and alpha**
>
> We chose the lambda according to the performance in the validation set. We will add the settings in the next version.
>
>  **Q: the explanation for the first increases of loos In figure 2(a)**
>
> We initialized the wight and bias of the last feed-forward layer as all zero, which makes the output distribution of teacher and student models become uniform distribution. Thus, in the beginning, the KL-divergence between student and teacher is zero. As the training progresses, the update of the teacher model is one step ahead of the student model and closer to the ground truth, which makes the discrepancy between the teacher model and the student model increase.
>
>   **Q: minor errors  in 4.3.1 and suggestions for detailed parameter number**
>
> Thanks for pointing out this error, and we have fixed it in the revision, and we will add the detailed parameter number in the next version.

---

### Official Review · AnonReviewer4 · 2020-10-26
**The proposed method is highly similar to some prior works. Experiments are not sufficient.**

**Rating:** 4
**Confidence:** 5

**Review:**

This paper follows the work of RCO[1], where knowledge distillation is conducted by learning from the optimization trajectories of the teacher rather than the converged teacher solely. The main difference is that in the proposed method ProKT, the student model learns from the teacher model step by step, while RCO downsamples the teacher's training trajectory with some sampling strategies, such as equal epoch interval strategy and greedy search strategy. The authors argue that smoothly changed targets can help the student out of poor local optima.

Pros:
1. Although some few works have made attempts to explore the training route of teacher for distillation, this research direction is still under-studied and of great potentials in my view.

2. The idea of making the teacher aware of the student for distillation is interesting.

Cons:
1. My main concern is that the proposed ProKT is highly similar to some prior works such as deep mutual learning [2], KDCL [3], and PCL [4] from the perspective of methodology, although their motivations are somewhat different. This paper lacks comparisons and clarifications of the differences between the proposed method and these prior methods.

2. The experiments are not sufficient to validate the proposed method. On image recognition tasks, comparisons are only conducted under heterogeneous architectures. However, distillation results under the same-style architecture should also be provided to give a more comprehensive view.

3. The proposed ProKT actually demands several times more computation resources than other methods like KD. Strictly speaking, evaluating the proposed method in accuracy only is not fair. How to make the proposed method more efficient, or how to make more fair comparisons should be carefully considered.

[1] Knowledge Distillation via Route Constrained Optimization, ICCV 2019.
[2] Deep mutual learning, CVPR 2018
[3] Online Knowledge Distillation via Collaborative Learning, CVPR 2020.
[4] Peer Collaborative Learning for Online Knowledge Distillation.

==============
post-rebuttal:
I have read all the comments from other reviewers and replies from the authors. All the reviewers are leaning to reject this paper due to the limited novelty and unfair and incomplete experimental comparisons. The authors' reply does not address my concerns, so I keep my initial attitude towards this work.

---

> ### Author Response · Authors · 2020-11-24
> **We sincerely thank you for your suggestions.**
>
> ### 1. Q: Relationship with deep mutual learning:
> A: We agree that the relationship between our proposed method and mutual learning needs to be further clarified. Moreover, the models involved in deep mutual learning settings typically have a similar capacity which differs from the capacity gap in the knowledge distillation settings. The key difference lies in the fact that though the learning procedure looks similar, Prokt has an asymmetric training objective: the student model does not get the supervision signal from the ground truth label while the teacher model does which corresponds to our motivation of gradient projection.
>
> ### 2. Q: Experiments in different settings (same-style architecture)
> A: Thanks for your suggestion. We will include the additional results on same-style architecture settings in the revised version.
>
> ### 3. Q：Fair evaluation on Prokt:
> A: We agree that the required additional computation source should be considered in the experiment design to make a fair comparison. We will try different implementation which includes the discretized version of Prokt and continuous version of RCO as reviewer 3 mentioned above.

---

### Official Review · AnonReviewer3 · 2020-10-26
**I vote for rejection as the paper lacks in novelty and results in small performance improvement.**

**Rating:** 3
**Confidence:** 5

**Review:**

**Summary**

This paper presents a method of knowledge distillation, which showed better results than KD and RCO in experimental results. The difference between this method, ProKT and RCO is in the teacher model. In this method, the snapshot interval of the teacher model targeted by the student is more frequent (every training step) than the RCO (authors called this property as *continuous*), and the parameters of the teacher model are also updated during *student* training (*dynamic*).

**Strong points**

- The proposed method was evaluated for two modalities (visual and textual). As far as I know, most KD papers evaluate only one modality (image classifier or language model). Experimenting with both modalities increases the experimental robustness of ProKT.

**Weak points**

- This method is not traditional KD, and the way the teacher model learns simultaneously with the student is called mutual learning, as the author briefly mentioned in the paper. There is a terminological problem that arises because of this: if the teacher model is also trained with the student, then neither model can be called a "teacher" or "student". Thus I think that most of the terms used in the paper need to be rewritten based on mutual learning terminology.
- In advance to the previous point, sharing knowledge between the big models and small models during the training is not new. To name a few, see [1, 2, 3].
- Aside from everything, the performance improvement that ProKT alone brings seems limited compared to CRD, and it is difficult to trust because there is no information such as the number of runs or standard deviation in the results of the experiment.

[1] Zhang, Ying, et al. "Deep mutual learning." Proceedings of the IEEE Conference on Computer Vision and Pattern Recognition. 2018.
[2] Anil, Rohan, et al. "Large scale distributed neural network training through online distillation." arXiv preprint arXiv:1804.03235 (2018).
[3] Park, Wonpyo, et al. "Diversified Mutual Learning for Deep Metric Learning." arXiv preprint arXiv:2009.04170 (2020).


**Recommendation**

I vote for rejection as the paper lacks in novelty and results in small performance improvement.

**Supporting arguments for the recommendation**

The recommendation is based on what was said in the weakness section. In addition, RCO does not provide any information from the student to the teacher during the training (just as Hinton KD or CRD does), so for a fair comparison, it seems necessary to compare the performance with RCO by separating only *continuous* facet among the two properties of ProKT (*continuous* and *dynamic*) claimed by the authors.

---

> ### Author Response · Authors · 2020-11-24
> **Thanks a lot for your constructive comments.**
>
> ### 1.Q:  The relationship between the proposed method and the mutual learning setting?
> A：We agree that the relationship between our proposed method and mutual learning needs to be further clarified. While our terminology selection is based on the ultimate goal of the training, i.e., whether the training procedure boosted one side or both sides. Moreover, the models involved in deep mutual learning settings typically have a similar capacity which differs from the capacity gap in the knowledge distillation settings.  The key difference lies in the fact that though the learning procedure looks similar, Prokt has an **asymmetric** training objective: the student model does not get the supervision signal from the ground truth label while the teacher model does which corresponds to our motivation of gradient projection.
>
> ### 2.Q: Experiment results of multiple runs and different settings:
> A: We have performed part of additional experiments on the Cifar-10 settings, and the results are consistent with our current claim. We will include the new results along with the experiments of continuous RCO in future submission.

---

### Official Review · AnonReviewer2 · 2020-10-28
**This paper proposes a novel and model agnostic knowledge distillation method which improves the accuracy of the student model.**

**Rating:** 5
**Confidence:** 3

**Review:**

The proposed method is simple yet effective and achieves state-of-the-art results on both image and text classification tasks. Compared to the existing approach (RCO and TA-learning), the main advantage is that ProKT avoids manual control of the learning process, which makes the ProKT more generally applicable. Important related works such as RCO and TA-learning are cited and compared. The paper does not contain a theory part, but wherever possible, equations are provided to illustrate how the method works.

Concerns:
From my understanding, ProKT is a continuous version of RCO with an extra constraint on the KL divergence of the probabilistic distributions of the teacher and the student models. As listed in Table 1, the accuracy improvement over RCO is marginal. The computational cost of ProKT is much higher than RCO as it requires training the teacher and student simultaneously. Therefore, I think the authors need to justify the benefits over RCO other than avoiding selecting the discrete learning objectives manually. From my understanding, ProKT is a continuous version of RCO with an extra constraint on the KL divergence of the probabilistic distributions of the teacher and the student models. As listed in Table 1, the accuracy improvement over RCO is marginal. The computational cost of ProKT is much higher than RCO as it requires training the teacher and student simultaneously. Therefore, I think the authors need to justify the benefits over RCO other than avoiding selecting the discrete learning objectives manually.

Moreover, this paper only provides an empirical evaluation on the CIFAR-100 dataset. It will be useful to see whether ProKT can achieve higher student accuracy on a more complex dataset such as ImageNet. The variability of results is also missing since there are no error bars for the results in Tables 1 and 2. Are the results averaged over multiple trials (if yes how many?), and is there a difference in variance between the methods?

In the proposed ProKT algorithm, both the student and teacher models are trained with the same number of iterations, and also the student model learns from an updated teacher model in each training step. Although the continuous and smooth learning targets might be desired,  I am curious about the performance of the possible variants of the ProKT algorithm. For example, let the teacher be several iterations ahead of the student or keep the same learning objective (teacher model) for several iterations, which can be viewed as something between ProKT and RCO.

Minor detail:
In Algorithm 1, it shows that the student model is updated before the teacher model. Should it be the teacher gets updated first?

Reasons for score: In general, I like the idea of turning the discrete learning objective into a continuous one. However, as the proposed algorithm is largely inspired by RCO, the novelty of the paper is incremental. Besides, the performance improvement of the proposed ProKT scheme over RCO also is marginal. I would consider raising my score if the authors could better demonstrate the advantage of ProKT over RCO.

---

> ### Author Response · Authors · 2020-11-24
> **Thanks a lot for your insightful comments.**
>
> Thanks a lot for your insightful comments. In the following, we will respond to each of your concerns and suggestions.
>
> ### 1. Q: the benefits over RCO
>
> Compared with RCO, our method has two benefits. In theory, our method distinguishes with RCO by the continuous target setting and the essential constraint in teacher training. The constraint provides a guarantee that the teacher model is projected in the neighborhood of student models. In the experiment, our improvement over RCO is consistent and reliable, especially in the heterogeneous setting, in which the capacity gap between the teacher model and the student model is significant while the projection process is specifically beneficial.
>
> Besides, the training time is not a bottleneck in practice. Because the model is trained once but runs unlimitedly, inference time is the main concern in the deployment of neural models. Our model has the same inference complexity as vanilla KD.
>
> ### 2. Q: suggestion on an experiment of ImageNet and adding the error bars.
> Thanks for your suggestions, we will conduct the experiment in ImageNet and add the error bars in the next version.
>
> ### 3. Q: the possible variants of the ProKT algorithm
> Thanks for your suggestions about more training variants. We will conduct the experiments and list the results in the next version.
>
> ### 4. Q: minor detail in Algorithm 1
> Thank you for pointing out that error in Algorithm 1. The teacher should get updated first. We fixed this error in the revision.

---

### Decision · Program_Chairs · 2021-01-07
**Final Decision**

**Decision:**

Reject

**Comment:**

This paper proposed a new variant of knowledge distillation. The basic idea is interesting although similar ideas have more or less appeared in the literature as pointed out by the reviewers. Our main concern on this work is that the real empirical improvements are too limited such that it is hard to conclude that the proposed method can really perform better than the baseline. In the meantime, the proposed method is much more computationally expensive.